# Suboptimal factors in maternal and newborn care for refugees: Lessons learned from perinatal audits in the Netherlands

**A. E. H. Verschuuren**[1]*, **J. B. Tankink**[2], **I. R. Postma**[1,3], **K. A. Bergman**[4], **B. Goodarzi**[5,6,7,8], **E. I. Feijen-de Jong**[5,8,9,10], **J. J. H. M. Erwich**[11]

**1** Department of Health Sciences, Global Health Unit, University Medical Centre Groningen & University of Groningen, Groningen, the Netherlands, **2** Erasmus University Medical Centre, Department of Obstetrics and Gynecology, Rotterdam, The Netherlands, **3** Department of Obstetrics and Gynecology, Isala Clinics, Zwolle, the Netherlands, **4** Department of Paediatrics Beatrix Children's Hospital, University Medical Centre Groningen & University of Groningen, Groningen, the Netherlands, **5** Department of Midwifery Science, Amsterdam UMC Location Vrije Universiteit Amsterdam, Amsterdam, The Netherlands, **6** Department of Primary Care and Longterm Care, University Medical Center Groningen & University of Groningen, Groningen, the Netherlands, **7** Amsterdam Public Health, Quality of Care, Amsterdam, The Netherlands, **8** Midwifery Academy Amsterdam Groningen, InHolland, Amsterdam, The Netherlands, **9** Department of Primary Care and Longterm Care, University Medical Centre Groningen & University of Groningen, Groningen, the Netherlands, **10** Midwifery Academy Amsterdam Groningen, InHolland, Groningen, the Netherlands, **11** Department of Obstetrics and Gynecology, University Medical Centre Groningen & University of Groningen, Groningen, the Netherlands

* a.e.h.verschuuren@umcg.nl

**Data Availability Statement:** Restrictions apply to the availability of these data because making the audit reports public would compromise patient

## Abstract

### Introduction

Refugees and their healthcare providers face numerous challenges in receiving and providing maternal and newborn care. Research exploring how these challenges are related to adverse perinatal and maternal outcomes is scarce. Therefore, this study aims to identify suboptimal factors in maternal and newborn care for asylum-seeking and refugee women and assess to what extent these factors may contribute to adverse pregnancy outcomes in the Netherlands.

### Methods

We conducted a retrospective analysis of national perinatal audit data from 2017 to 2019. Our analysis encompassed cases with adverse perinatal and maternal outcomes in women with a refugee background (n = 53). Suboptimal factors in care were identified and categorized according to Binder et al.'s Three Delays Model, and the extent to which they contributed to the adverse outcome was evaluated.

### Results

We identified 29 suboptimal factors, of which seven were related to care-seeking, six to the accessibility of services, and 16 to the quality of care. All 53 cases contained suboptimal factors, and in 67.9% of cases, at least one of these factors most likely or probably contributed to the adverse perinatal or maternal outcome.

privacy. We obtained data from Perined, which is a Dutch organization that facilitates the national perinatal registry and audit databases. Parties such as healthcare providers and researchers seeking access to the unprocessed data can submit a formal request through Perined's designated data request form. For more information contact info@perined.nl.

**Funding:** This study was supported by the research fund of the medical center Leeuwarden, the Netherlands. Grant number: 2021-13. The funders had no role in study design, data collection and analysis, decision to publish, or preparation of the manuscript.

**Competing interests:** The authors declare no conflict of interest. The sponsors had no role in the design, execution, interpretation, or writing of the study.

## Conclusion

The number of suboptimal factors identified in this study and the extent to which they contributed to adverse perinatal and maternal outcomes among refugee women is alarming. The wide range of suboptimal factors identified provides considerable scope for improvement of maternal and newborn care for refugee populations. These findings also highlight the importance of including refugee women in perinatal audits as it is essential for healthcare providers to better understand the factors associated with adverse outcomes to improve the quality of care. Adjustments to improve care for refugees could include culturally sensitive education for healthcare providers, increased workforce diversity, minimizing the relocation of asylum seekers, and permanent reimbursement of professional interpreter costs.

## Introduction

The rise of forced migration worldwide urges maternal and newborn care providers to respond to the needs of pregnant refugees and their children [1]. In the Netherlands, approximately 600 babies are born annually to women living in asylum seeker centers. The number of babies born to refugee women with a residence permit is likely higher, but the exact number remains unknown [2]. A substantial body of international literature has demonstrated that asylum seekers and refugee women with residence permits more often experience adverse perinatal and maternal outcomes compared to non-migrant populations, including higher rates of perinatal and maternal mortality and morbidity [2–6]. In the Netherlands, one study showed a seven times higher perinatal mortality among recently arrived asylum seekers compared to the local Dutch population [4].

Given these inequities, access to high-quality maternal and newborn care is essential to promote the health and well-being of pregnant refugees. However, these women face many challenges in accessing maternal and newborn care, such as linguistic differences, disadvantaged socio-economic status, racial, ethnic, and cultural discrimination, limited knowledge of the host country's healthcare system, and the stress of resettlement in a new country [7–9]. Maternal and newborn care providers also face numerous challenges in providing care to refugees [10, 11]. A previous study on maternal and newborn care for refugees in the Netherlands identified five themes of challenges community care midwives face while providing care for refugee women: interdisciplinary collaboration, communication with clients, continuity of care, psychosocial care, and the vulnerable context of clients [12].

The effect these challenges have on the quality of maternal and newborn care and their direct or indirect association with adverse perinatal and maternal outcomes remain poorly studied in refugee populations. Our study aims to fill this gap by identifying suboptimal care factors and evaluating to what extent these factors contribute to adverse perinatal and maternal outcomes among refugee women in the Netherlands. Based on the identified suboptimal care factors, we will formulate recommendations for policy and practice in maternal and newborn care for refugee women.

Research question: Which suboptimal factors play a role in maternal and newborn care for refugees, and to what extent do they contribute to adverse perinatal and maternal outcomes?

## Methods

### Design

We conducted a retrospective audit of cases from the Dutch National Perinatal Audit registry, which concerned adverse perinatal and maternal outcomes in refugee women over three years (2017–2019).

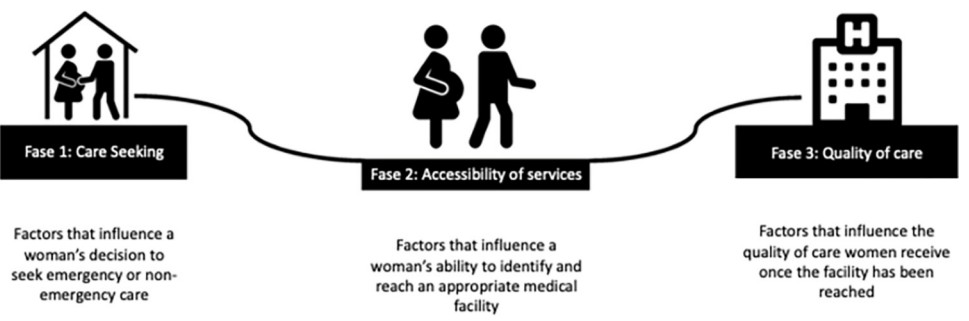

**Fig 1. The Three Delays Model** [15, 16].

## The perinatal audit registry

Local perinatal audits in the Netherlands are confidential enquiries of severe maternal or perinatal morbidity or mortality cases. These audits take place in all maternal and newborn care centers in the Netherlands. During an audit meeting, healthcare professionals systematically review case reports and identify improvement and action points to enhance practice. For every case reviewed, a detailed case report is stored, documenting case characteristics and details about the perinatal care provided. The selection of cases reviewed depends on the submissions from healthcare providers and whether the cases align with one of the four audit themes. During the course of this study, the audit themes, chosen on a national level as focal points, encompassed late premature mortality (occurring between 32+0 and 36+6 weeks of pregnancy), perinatal asphyxia (occurring above 37+0 weeks of pregnancy), hyperbilirubinemia, and uterine rupture. The Dutch National Perinatal Audit registry is a comprehensive database containing all cases reviewed in regional perinatal audits in the Netherlands [13]. A more detailed explanation of the Perinatal Audit registry is included in Appendix 1 in S1 File.

## Theoretical framework: The Three Delays Model

In 1994, Thaddeus and Maine proposed the Three Delays Model to facilitate the identification of factors that cause a delay in care and might therefore contribute to adverse outcomes [14]. The model identifies three phases of possible delay (see Fig 1).

The model was originally designed for low-resource settings but was modified by Binder et al. to evaluate maternal and newborn care for migrant populations in high-income settings [16]. In this study, we employ the modified version of the model to categorize suboptimal factors of care.

## Study population

We included all cases that concerned women with a refugee background included in the Dutch National Perinatal Audit registry. Women with a refugee background were defined as asylum-seeking women, women with recently obtained residence permits, and undocumented women, altogether referred to as 'refugees' in this study.

To identify refugee women in the Perinatal Audit registry, the first author hand-searched all case reports between March and May 2021 as migration history and legal status are not routinely included in the Perinatal Audit registry's administration. Women's cases were deemed eligible for inclusion if either the country of origin or terminology in the case report indicated a refugee background (see Box 1). The determination of which countries of origin rendered women eligible for inclusion depended on both the number of asylum applications and the

---

**Box 1. Words, phrases, and countries of origin encountered in reports, which made women's cases eligible for inclusion**

**Words or phrases encountered in case reports which made them eligible for inclusion:** Asylum seeker; Asylum seekers center; Refugee camp; Refugee status; Residence permit; Fled from; Dutch Council for Refugees; GCA/GZA*; Temporary residence in the Netherlands due to political tensions back home.

Countries of origin encountered in case reports which made them eligible for inclusion:

Syria; Somalia; Iran; Eritrea; Afghanistan; Pakistan; Middle East; Congo; Ethiopia; Turkey**; Ghana; Nigeria.

* Organization that provides primary healthcare for asylum seekers in the Netherlands.

** Only eligible for inclusion in combination with another factor such as a short stay in the Netherlands

---

percentage of immigrants that applied for asylum from these countries, as reported by Statistics Netherlands [17]. Cases were excluded if women had lived in the Netherlands for more than ten years, if there was uncertainty about a woman's migration background, or if there was no record of the provided care.

## Identification and classification of suboptimal factors

After case selection, we identified suboptimal factors by independently reviewing all case reports. This was done by the first two authors in collaboration with an expert team, that consisted of a midwife (EF-dJ), an obstetrician (JE), an obstetrician in training (IP), and a neonatologist (KB). The experts considered care suboptimal for patient-related factors (phases one and two of the Three Delays Model) if they negatively affected the care refugee women received with a possible negative effect on the outcome. Quality-related factors (phase three of the Three Delays Model) were considered suboptimal if care deviated from the professional requirements of standard care, national guidelines, or local protocols. Cases in which there was no consensus on suboptimal factors were discussed in a meeting with the entire team and conflicts were resolved. After the identification of suboptimal factors, we classified them according to Binder et al.'s Three Delays Model [16].

We conducted this analysis in addition to the data available from the original audit as our objective was to also identify patient-related suboptimal factors that were not included in original audits. During all stages of the analysis, the research team was blinded to the suboptimal factors identified in original audits. Additional suboptimal factors identified by the original audits were included after classification. We conducted descriptive statistics on suboptimal factors using SPSS version 28.0.0.0.

## Minor/Major analysis

After the identification of suboptimal care factors, we assessed the likelihood that a suboptimal factor contributed to the adverse outcome, by labeling factors either 'minor' or major'. Factors were considered 'minor' if any contribution to the adverse outcome was unlikely or uncertain. Factors were considered 'major' if they most likely or probably contributed to the adverse outcome. These decisions were made by the expert team and based on their professional judgment.

### Ethics

This study was assessed by the medical ethical committee of the University Medical Centre Groningen (METc 2021/375) and was not subject to the Medical Research Involving Human Subjects Act in the Netherlands. With regards to privacy, the researchers exclusively accessed anonymized data contained within a secured file. So all data were retrieved and handled anonymously. Perined, the organization responsible for administering and maintaining the National perinatal audit, provided authorization for this research, and conducted a thorough review of the manuscript, approving the final version for publication.

## Results

### Selection of cases

Of all 1117 cases stored in the national Perinatal Audit, in 53 (4.7%) cases women were identified as refugees. Most cases (n = 33, 62.3%) were included because women originated from countries that suggested a forced migration background. The other 20 cases (37.7%) were included because the terminology in the case report suggested that the women had a refugee background (see Box 1).

### The audit processes

During the audit process, we discussed seventeen cases with all members of the research team due to doubt from a team member whether a factor should be classified as suboptimal. In a three-hour-long meeting, all members of the team discussed the cases and two general issues, which included the definition of an untimely start of antenatal care and how to handle missing data in case reports. During this meeting, all discrepancies and unclarities between experts were resolved. After the team reached a consensus, suboptimal factors were finalized. A detailed description of the suboptimal factors is included in Appendix 2 in S1 File.

After completing our analysis, we added the descriptions of suboptimal care recorded in the national perinatal audit database from the original audits. These original audits described 119 suboptimal care factors, divided over 43 cases. Ten cases contained no suboptimal factors in the Perinatal Audit Registry. Of the suboptimal factors identified in the original audits, 45.4% (n = 54) addressed the same suboptimal care identified by the research team, while 45.4% (n = 54) were not identified by the research team's analysis. All additional factors corresponded to our framework and were grouped into 29 suboptimal factor categories. Of the suboptimal factors identified by original audits 8.4% (n = 10) were not included because they concerned team evaluations or peer support for healthcare providers (n = 4), it was unclear what specific suboptimal factor they targeted (n = 3) or it was unclear what was meant (n = 3).

### Characteristics of included cases

Table 1 shows the characteristics of the included cases. Women were born in Asia, Africa, or Europe. At the start of pregnancy care, 24.5% of women were asylum seekers and 22.6% were refugees with a residence permit. In the rest of the cases (50.9%), women's residence status was missing from the perinatal audit data.

### Adverse outcomes

Adverse outcomes from cases were divided into five categories: fetal death (n = 14), perinatal asphyxia (n = 15), severe neonatal hyperbilirubinemia (n = 12), uterine rupture (n = 7), and other (n = 7). The category 'other' included neonatal mortality (n = 2), postpartum hemorrhage (n = 2), pre-eclampsia (n = 1), meconium aspiration syndrome (n = 1), GBS-sepsis (n = 1) and one case in which a woman suffered from pre-eclampsia, placental rupture, and

**Table 1. Case characteristics.**

| Case characteristics | Total (N = 53) |
|---|---|
| **Origin of the mother** | |
| Asia* | 26 (49.1) |
| Africa† | 20 (37.7) |
| Europe‡ | 3 (5.7) |
| Unknown | 4 (7.5) |
| **Residence status at the start of pregnancy care** | |
| Asylum seeker | 13 (24.5) |
| Refugee with a residence permit | 12 (22.6) |
| Unknown | 28 (52.8) |
| **Duration of stay in the Netherlands (years)** | |
| < 1 year | 18 (34.0) |
| < 2 years | 6 (11.3) |
| 3–4 years | 9 (17.0) |
| 4–10 years | 6 (11.3) |
| Unknown | 14 (26.4) |
| **Age** | |
| < 20 | 4 (7.5) |
| 20–29 | 20 (37.7) |
| 30–39 | 26 (49.1) |
| 40+ | 3 (5.7) |
| **Parity** | |
| Nulliparous | 15 (28.3) |
| Multiparous (1,2,3) | 32 (60.4) |
| Grand multipara (≥4) | 6 (11.3) |

Data are presented as: Number of cases (%)

* Asian countries included: Syria, Iran, Iraq, Afghanistan, Pakistan, and Turkey

† African countries included: Somalia, Eritrea, Nigeria, Ethiopia, Congo, Ghana, Sudan, and Gambia

‡ European countries included: Bosnia, Belarus, and Moldavia

postpartum hemorrhage (n = 1). The number of suboptimal factors per adverse outcome is included in Appendix 3 in S1 File.

## Suboptimal factors

We identified 29 suboptimal factor categories, which are grouped according to the Three Delays Model in Table 2. Seven suboptimal factors were related to care-seeking (first delay), six to the accessibility of services (second delay), and sixteen to the quality of care (third delay). In 67.9% of cases (n = 36) at least one suboptimal factor most likely or probably contributed to the adverse outcome. Most of these major suboptimal factors occurred in phase three of the three delays model, followed by phase one and phase two. The number of cases with major suboptimal factors in phase three is especially high in cases of severe neonatal hyperbilirubine-mia (11 out of 12 cases, 91.7%).

## Phase one: Care seeking

Suboptimal factors with a possible effect on care seeking occurred in 43 cases (81.1%), and in fourteen cases (26.4%) at least one of these factors most likely or probably contributed to the

**Table 2. Suboptimal factors and their association with adverse outcomes (number of major factors) grouped by phase of delay.**

| Suboptimal factors | Number of cases in which suboptimal care was present, n (%) | Number of times factors were labeled major |
|---|---|---|
| **Total** | **53** | |
| **Phase 1: Care seeking** | **43 (81.1)** | **14** |
| Untimely start of antenatal care | 22 (41.5) | 1 |
| Missed appointments/late arrival | 22 (41.5) | 3 |
| Non-compliance | 20 (37.7) | 3 |
| Misunderstanding | 10 | 0 |
| Patient's choice | 2 | 1 |
| Unclear | 10 | 2 |
| Delayed care seeking in case of alarm symptoms | 18 (34.0) | 7 |
| Vulnerable context | 15 (28.3) | 1 |
| Partially unmonitored pregnancy | 5 (9.4) | 0 |
| Lack of trust in healthcare provider | 2 (3.8) | 1 |
| **Phase 2: Accessibility of services** | **50 (94.3)** | **8** |
| Language barrier | 45 (84.9) | 7 |
| Inadequate involvement of an official interpreter | 31 (58.5) | 7 |
| Transportation difficulties | 12 (22.6) | 1 |
| Transfer of care | 10 (18.9) | 0 |
| Financial barriers | 3 (5.7) | 0 |
| Uncertainty or stress surrounding the asylum procedure | 3 (5.7) | 0 |
| **Phase 3: Quality of care** | **53 (100)** | **29** |
| Communication issues between care providers | 33 (62.3) | 4 |
| Missed or late diagnostic tests | 33 (62.3) | 13 |
| Other inadequate care | 25 (47.2) | 4 |
| No or late start of treatment | 24 (45.3) | 14 |
| Incomplete history taking or counseling | 24 (45.3) | 7 |
| Issues concerning documentation | 19 (35.8) | 0 |
| Missed or late diagnosis | 18 (34.0) | 13 |
| Logistic or technical issues | 16 (30.2) | 2 |
| Delay in consultation or referral | 16 (30.2) | 7 |
| Insufficient or inadequate psychosocial care | 14 (26.4) | 0 |
| Inadequate action in case of no-show | 8 (15.1) | 0 |
| Healthcare providers' negative attitude | 8 (15.1) | 0 |
| Insufficient monitoring during labor | 7 (13.2) | 2 |
| Issues with postnatal maternity care | 6 (11.3) | 2 |
| Inadequate risk assessment | 4 (7.5) | 1 |
| No placental pathology while indicated | 4 (7.5) | 0 |

A detailed description of suboptimal factors can be found in Appendix 2 in S1 File

adverse outcome (labeled as major)The most common suboptimal factors were an untimely start of antenatal care (n = 22), missed or late arrival at appointments (n = 22), and non-compliance (n = 20) (see Table 2). Of all suboptimal factors in phase one, delayed care seeking most often contributed to the adverse outcomes (n = 7, 13.2%). Case A presents an example, in which major contributing factors in phase 1 were missed appointments and delayed care seeking. In phase two, a language barrier and inadequate involvement of an official interpreter were major suboptimal factors. The major factor identified in phase three was missed or late diagnostic tests.

**Case A.**   A young multiparous mother from the Middle East, who had been in the Netherlands as an asylum seeker for less than a year, frequently missed appointments throughout her pregnancy. Due to miscommunication, the patient missed blood glucose measurements and didn't go to a lab appointment her midwife had scheduled her for. Healthcare providers mentioned a language barrier as the reason for miscommunication and the patient's missed appointments. At 32 weeks of pregnancy, the patient was referred to the hospital because her community care midwife suspected fetal growth restriction. Due to another miscommunication, the patient did not show up at the ultrasound appointment in the hospital. After three weeks, her midwife arranged a new appointment, and fetal growth restriction was diagnosed. The obstetrician decided that the fetal growth ultrasound must be repeated after two weeks, even though an additional ultrasound for doppler-flow measurements after one week would have been indicated according to Dutch care guidelines. More than two weeks later, with no record of a new fetal growth ultrasound, the patient's partner phoned the hospital with signs of labor. After arrival at the hospital, healthcare providers found no fetal heartbeat and fetal death was diagnosed. When asked, the patient reported that she hadn't felt any fetal movements in the two days before the hospital visit.

## Phase two: Accessibility of care

Suboptimal factors with a possible effect on the accessibility of care occurred in 50 cases (94.3%). In eight cases (15.1%) at least one of these factors most likely or probably contributed to the adverse outcome (labeled as major). The most common suboptimal factors for refugee women while accessing perinatal care were language barriers (n = 45) and inadequate involvement of an official interpreter (n = 31) (see Table 2). In seven cases (13.2%), these factors most likely or probably contributed to an adverse outcome. Case B presents an example, in which the major contributing factor for phase one concerned missed appointments. The major contributing factors in phase two were a language barrier and inadequate involvement of an official interpreter and in phase three this was an issue with postnatal maternity care.

**Case B.**   A primiparous woman from Afrika, who had been in the Netherlands for a little over a year, had an uncomplicated pregnancy when labor started after 41 weeks of gestation. The patient did not speak any Dutch or English. Her partner served as an interpreter, but his Dutch language skills were limited. After a difficult labor, complicated by shoulder dystocia, the patient gave birth to a child with a suboptimal start who recovered quickly (Apgar score of 8 after 1 minute, and 9 after 5 minutes). That same evening at nine pm, the family was discharged from the hospital, and maternity care services were called for a home visit. As the concept of maternity care services was not sufficiently explained to the woman and her partner, they were asleep and had their phones turned off when the maternity care assistant rang the door of their home that evening and the next morning. When the community midwife arrived later that day, she discovered that the newborn's temperature had not been monitored, there were no hot water bottles in the baby's bed, and breastfeeding had not yet succeeded. Moreover, the baby looked yellow. The midwife immediately arranged hospital admission, where healthcare providers started treatment with phototherapy for hyperbilirubinemia.

## Phase three: Quality of care

Suboptimal factors with a possible effect on quality of care occurred in 53 cases (100%), and in 29 cases (54.7%) at least one of these factors most likely or probably contributed to the adverse outcome (labeled as major). Suboptimal factors that most often contributed to adverse outcomes were no or late start of treatment (n = 14, 26.4%), missed or late diagnosis (n = 13, 24.5%), and missed or late diagnostic tests (n = 13, 24.5%). As a part of missed or late

diagnostic tests, late diagnostics after detecting neonatal jaundice occurred in seven cases and always most likely or probably contributed to the adverse outcome (labeled as 'major' factors). Communication issues between care providers were the most common suboptimal factor (n = 33, 62.3%), although its contribution to the adverse outcome was often unlikely or uncertain (n = 29, 54.7%). A negative attitude by healthcare providers in case reports was observed in eight cases and included describing patients as 'incapable of following instructions', 'unreasonable', 'uncooperative', and 'unmanageable'. In all these cases communication issues had been described during pregnancy. Case C presents an example in which all suboptimal factors in phases one and two were deemed minor. Phase three was assessed major for incomplete history taking, late diagnostic tests, missed diagnosis, and a delay in referral.

**Case C.**   A primiparous woman from the Middle East had been in the Netherlands as an asylum seeker for around 6 months. She experiences an uncomplicated pregnancy and childbirth. During pregnancy, healthcare providers did not assess risk factors for neonatal hyperbilirubinemia while family history would have revealed a high risk. Two days postpartum, the newborn's skin and eyes appeared yellow, and the child had lost 9% of its birth weight. Two days later, even though the newborn's weight had increased 80 grams, the skin was still yellow, and the maternity care assistant discovered urate crystals in the urine. No action was undertaken by any of the care providers. Seven days postpartum, the mother expressed worry because her baby hadn't defecated for three days and seemed less alert. The midwife immediately referred the family to the hospital where the baby was diagnosed with severe hyperbilirubinemia (bilirubin: 389 umol/l) and treated with phototherapy.

## Discussion

The aim of this study was to identify suboptimal factors in maternal and newborn care for refugee women and assess to what extent these factors contribute to adverse pregnancy outcomes in the Netherlands. Our results indicate that in 67.9% of cases, at least one suboptimal factor most likely or probably contributed to the adverse perinatal or maternal outcome. We identified 29 suboptimal factor categories, which were categorized according to the Three Delays Model. Seven factors were related to care-seeking (1st delay), six to the accessibility of services (2nd delay), and sixteen to the quality of care (3rd delay). Most of the suboptimal factors in maternal and newborn care identified in this study have been previously reported in refugee populations [4, 5, 10, 12, 18–30]. Previous studies show that suboptimal maternity care is more prevalent among refugee women compared to non-migrant populations [19, 20]. In our study, we explicitly demonstrate that suboptimal factors in maternal and newborn care contributed to adverse perinatal and maternal outcomes among refugees. Our findings in this context thus imply that suboptimal factors in maternal and newborn care play a role in perinatal health inequities between refugee and non-migrant populations, highlighting the need for targeted interventions in this area.

The wide variety of suboptimal factors identified in this study and their association with adverse outcomes challenge the Dutch healthcare system's fundamental principles of access to care, equity, and high-quality services for all [31]. Moreover, forced migration into the Netherlands has a long-standing history, and health inequities have disadvantaged various migrant populations for decades [6, 32, 33]. The subsequent paragraphs will discuss suboptimal factors identified in this study per phase of delay in care and provide recommendations on how to address these factors to improve maternal and newborn care for refugees.

### Phase 1: Care-seeking

The suboptimal factors related to care seeking in this study, emphasize the importance of promoting and facilitating care-seeking behavior among refugee women. However, it is important

to recognize that limited care-seeking and perceived non-compliance are not solely attributable to refugees, but frequently stem from structural barriers on the individual, healthcare service, and migration policy levels [34–36]. To improve refugee women's access to care, it is crucial to acknowledge that addressing individual behaviors alone is insufficient and that interventions must also target underlying structural barriers, such as socio-economic disadvantage, women's educational attainment, unwelcoming attitudes towards refugees, and healthcare providers' lack of cultural competence [5, 27, 34–38]. Further research should explore the extent to which different structural barriers affect refugee women's access to care and identify best practices in this regard. In addition, Policy maker and healthcare providers should collaborate with refugees to develop and implement future interventions and research [39–42].

## Phase 2: Accessibility of services

The most common suboptimal factor in phase 2, which also most frequently contributed to the adverse outcomes in this phase, was a language barrier. These results add to a large body of evidence demonstrating the harmful impact of unaddressed language barriers in healthcare [5, 26, 27]. In many of the audited cases, official interpreters were not routinely involved, and care providers commonly relied on women's language skills, nonverbal communication, or informal interpreters. These alternative communication strategies limit women's ability to understand medical information and compromise the safety, confidentiality, and accuracy of translations [43, 44]. Barriers to language support can be a direct consequence of political choices. For instance, during our study period community care midwives in the Netherlands were unable to claim the costs of professional interpreter services for refugees with a residence permit while before 2012 these services were freely available for healthcare providers. This stresses the need for the permanent reimbursement of interpreter costs in all refugee receiving countries, which the Dutch government reinstalled in maternal and newborn care as of January 2023 [45]. Although studies show that the presence of professional interpreters improves clinical outcomes and patient satisfaction with care, professional interpreters alone do not resolve all communication barriers [43, 46, 47]. Other factors, such as cultural differences, women's educational attainment and experiences of discrimination or stigma, also influence communication in maternal and newborn care for refugees [5, 12, 38]. Thus, to overcome communication barriers, efforts towards an inclusive healthcare system should be made, which encompasses culturally sensitive care that considers the widely diverse backgrounds of refugees [48]. One of the initial steps in realizing this objective is to train both current and aspiring healthcare professionals in cultural humility [49–52].

Other factors leading to phase 2 delays, such as transfer of care and stress surrounding the asylum procedure, illustrate how migration policy and the asylum seeker context compromise women's ability to access care [4, 12, 35, 53]. Transfer of care often occurs due to the relocation of asylum seekers and leads to partially uncontrolled pregnancies, missed appointments, and missed or repeated diagnostic tests. These findings add to a growing body of evidence on the negative effects of relocations on the well-being of pregnant asylum seekers as well as the continuity and quality of care [4, 12, 18, 28–30]. This calls for an adjustment to the Dutch refugee system which limits the number of relocations for all asylum seekers, especially during pregnancy.

## Phase 3: Quality of care

While several suboptimal factors observed in phase 3 have been reported in non-refugee populations, we also identified factors that appear more specific to refugees. These concern incomplete history taking or inadequate counseling, particularly regarding prenatal diagnostics, and

issues with post-partum care, such as delayed arrangement [10, 19–24, 54, 55]. Furthermore, in contrast with previous audit studies that did not focus on refugees specifically, our findings present evidence for negative attitudes among healthcare providers in care for refugee women [10, 19–24]. Previous research on refugee women's experiences and healthcare staff's attitudes shows that racial and ethnic discrimination in care is common [5, 56–58]. This is concerning, as racism adversely affects the quality of care refugee women receive and is associated with a lack of trust and delayed care seeking [59–68]. In many cases, healthcare providers may be unaware of their discriminatory behavior, as it may result from unconscious bias, prejudices, or stereotyping [69]. Further research is necessary to better understand how implicit bias and discrimination affect the quality of maternity care provided to refugee populations in the Netherlands. In addition, efforts should be made to increase workforce diversity, as cohorts of both current and training healthcare providers are often not representative of the populations they serve [70, 71]. This is of fundamental importance as workforce diversity improves the cultural sensitivity of care and is associated with improved patient satisfaction, better communication between patients and their healthcare providers and a reduced risk of severe maternal outcomes [50, 51, 70–74].

## Strengths and limitations

This study adds to the existing literature by providing a more in-depth presentation of suboptimal factors in maternal and newborn care for refugees and examining which suboptimal factors contribute to adverse perinatal and maternal outcomes. The unique database which allowed us to describe suboptimal factors in great detail and assess to what extent they contribute to adverse outcomes poses a major strength of this study. Additionally, the Three Delays Model offers a comprehensive framework for understanding how delays in care can contribute to adverse outcomes and provides valuable insights for developing targeted interventions. The involvement of experts from all care professions involved in maternal and newborn care for refugees and the unanimous consensus reached on suboptimal factors by these experts strengthen the validity of our conclusions.

The main limitation is that the Perinatal Audit registry only includes cases discussed in local audits and therefore presented a selection of cases with only adverse pregnancy outcomes. Due to the explorative scope of the study, we decided not to compare suboptimal factors between refugee and non-refugee populations, which limits conclusions on population-specific factors that influence care. In addition, reports stored in the Perinatal Audit registry contain summaries of patient records, which can make it challenging to distinguish inadequate documentation from actual instances of suboptimal care. To tackle this problem, we did not assign a suboptimal factor if there was any unclarity on whether or how care was provided. This probably explains why 45.4% of the suboptimal factors identified in the original local audits were not identified by our research team. Our outcomes therefore most likely reflect an underrepresentation of suboptimal factors in the study population.

## Conclusion

The number of suboptimal factors identified in this study and the extent to which they contributed to adverse perinatal and maternal outcomes among refugee women is alarming. The wide range of suboptimal factors identified provides considerable scope for improvement of maternal and newborn care for refugee populations. These findings also highlight the importance of including refugee women in perinatal audits as it is essential for healthcare providers to better understand the factors associated with adverse outcomes to improve the quality of care. To improve care for refugees initiatives such as culturally sensitive education for healthcare

providers, increased workforce diversity, minimizing the relocation of asylum seekers, and permanent reimbursement of professional interpreter costs are necessary.

## Supporting information

**S1 File.**
(DOCX)

## Acknowledgments

The authors wish to thank A. Schonewille-Rosman from Perined, who enabled the audit database search and approved the final manuscript for publication.

## Author Contributions

**Conceptualization:** A. E. H. Verschuuren, J. B. Tankink, I. R. Postma, E. I. Feijen-de Jong, J. J. H. M. Erwich.

**Data curation:** A. E. H. Verschuuren, J. J. H. M. Erwich.

**Formal analysis:** A. E. H. Verschuuren, J. B. Tankink, I. R. Postma, K. A. Bergman, E. I. Feijen-de Jong, J. J. H. M. Erwich.

**Funding acquisition:** A. E. H. Verschuuren, I. R. Postma, E. I. Feijen-de Jong.

**Investigation:** A. E. H. Verschuuren, I. R. Postma, J. J. H. M. Erwich.

**Methodology:** A. E. H. Verschuuren, J. B. Tankink, I. R. Postma, E. I. Feijen-de Jong, J. J. H. M. Erwich.

**Supervision:** I. R. Postma, B. Goodarzi, E. I. Feijen-de Jong, J. J. H. M. Erwich.

**Visualization:** A. E. H. Verschuuren.

**Writing – original draft:** A. E. H. Verschuuren.

**Writing – review & editing:** A. E. H. Verschuuren, J. B. Tankink, I. R. Postma, K. A. Bergman, B. Goodarzi, E. I. Feijen-de Jong, J. J. H. M. Erwich.

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
