## [Decision Letter · Decision Letter 0]

4 Dec 2023

PONE-D-23-31690Suboptimal factors in maternal and newborn care for refugees: lessons learned from perinatal audits in the Netherlands.PLOS ONE

Dear Dr. Verschuuren,

Thank you for submitting your manuscript to PLOS ONE. After careful consideration, we feel that it has merit but does not fully meet PLOS ONE’s publication criteria as it currently stands. Therefore, we invite you to submit a revised version of the manuscript that addresses the points raised during the review process.

**Major revisions required**==============================

We look forward to receiving your revised manuscript.

Kind regards,

Omid Dadras, MD, PhD

Academic Editor

PLOS ONE

“This study was supported by the research fund of the medical center Leeuwarden, the Netherlands. Grant number: 2021-13.”

6. We notice that your supplementary tables are included in the manuscript file. Please remove them and upload them with the file type 'Supporting Information'. Please ensure that each Supporting Information file has a legend listed in the manuscript after the references list.

Reviewers' comments:

Reviewer's Responses to Questions

**Comments to the Author**

1. Is the manuscript technically sound, and do the data support the conclusions?

Reviewer #1: No

Reviewer #2: Partly

Reviewer #3: Partly

2. Has the statistical analysis been performed appropriately and rigorously? 

Reviewer #1: I Don't Know

Reviewer #2: N/A

Reviewer #3: Yes

3. Have the authors made all data underlying the findings in their manuscript fully available?

Reviewer #1: Yes

Reviewer #2: Yes

Reviewer #3: No

4. Is the manuscript presented in an intelligible fashion and written in standard English?

Reviewer #1: No

Reviewer #2: Yes

Reviewer #3: Yes

5. Review Comments to the Author

Reviewer #1: Dear Author,

I found your paper interesting; however, I have a few comments and suggestions for improvement

Clarity and Flow:

Work on simplifying language and ensure a smooth flow.

Collaborate with team members for input before finalizing.

Table Presentation:

Enhance the visual appeal of the table for clarity.

Ensure that the data selected is both relevant and effective.

Visualization:

Consider presenting findings in diagrammatic form for a more engaging presentation.

Literature Review:

Expand the literature review to include more suboptimal factors in the discussio.

Avoid singular focus and explore a comprehensive range of factors such as education, financial costs, health insurance, wait times, age, and gender in the context of access to healthcare.

Ensure clarity and conciseness in presenting the expanded list of factors for a more comprehensive understanding.

Conciseness:

Make the paper more clear and concise, avoiding unnecessary details.

List and present factors explicitly to enhance readability.

By addressing these points, the paper can become more accessible, visually appealing, and comprehensive.

Reviewer #2: The author has written the overall manuscript well. However, the sample size and conclusion are dubious.

Line 209.

Pakistan and Afghanistan are not Middle Eastern countries. I suggest the authors revise the tables and findings.

Line 260.

Table 3.

Postnatal maternity care should incorporate dietary considerations, including religious restrictions, for refugees. It is pertinent to evaluate why dietary factors are not addressed in the table 3 during institutional or non-institutional-based care, particularly given the focus of the study on refugees. The lack of attention to this aspect may lead to suboptimal care and outcomes for refugee women, who may have unique dietary needs due to religious restrictions or other factors.

Line 408-409

The overall conclusion appears to be too general in nature. Specifically, the authors should direct their attention towards addressing the unique needs of refugee women in the Netherlands. A more targeted approach would be more effective in delivering the intended message. A more targeted approach would be more effective in delivering the intended message.

Reviewer #3: In the throes of the world's current multiple wars, within and between many countries, the authors' research is a timely and significant contribution towards global best practice for managing the welfare of refugees, in particular, pregnancy and perinatal care.

Their data analysis appears appropriate for a reviewer with limited statistical expertise that I am.

However, their findings do not support the generalizability reflected in their results and conclusions, and their "Strengths and Limitations" section partly reveals this point. There are methodological limitations including retrospective design, using a national audit whose cases are incomplete, with an unknown denominator, subjective criteria for case inclusion/rejection by the research team, absence of non-refugee population data for background and comparison.

In my opinion, 'minor revision' is warranted such that the manuscript would meet acceptance for publication. Its conclusions and recommended policy adjustments ought to be limited to the scope of the audit, namely, the findings that need future corroboration to drive policy change.

6. PLOS authors have the option to publish the peer review history of their article (what does this mean?). If published, this will include your full peer review and any attached files.

Reviewer #1: **Yes: **Rita Adhikari

Reviewer #2: **Yes: **Ishtiaq Ahmad

Reviewer #3: No

---

## [Author Response · Author response to Decision Letter 0]

28 Apr 2024

Author’s response: 

Reviewer #1 

Dear reviewer, 

Thank you for your constructive, relevant, and helpful comments to improve the manuscript. We used the track changes tool to indicate the edits we made. 

Comment 1: 

Dear Author, 

I found your paper interesting; however, I have a few comments and suggestions for improvement 

Response 1: 

Thank you for your comments and suggestions. We hope to have answered them to your satisfaction. 

Comment 2: 

Clarity and Flow: 

Work on simplifying language and ensure a smooth flow. 

Collaborate with team members for input before finalizing. 

Response 2: 

After we changed the manuscript according to all reviewer’s comments, all authors thoroughly reviewed the manuscript to improve language and readability. 

All changes made based on reviewers’ comments were approved by the entire research team before re-submission. 

Comment 3: 

Table Presentation: 

Enhance the visual appeal of the table for clarity. 

Ensure that the data selected is both relevant and effective. 

Response 3: 

We decided to remove Table 2 entirely as we agree with the reviewer that it was difficult to interpret and added relevant information to the text. We adjusted the text accordingly. In addition, to improve clarity and readability we removed the number of minor suboptimal factors from table 3. The visual appeal of the table will be further amplified by the journal before publication. 

Comment 4: 

Visualization: 

Consider presenting findings in diagrammatic form for a more engaging presentation. 

Response 4: 

We presented the Three Delays Model in a diagrammatic format as suggested in the comments in the manuscript, to clarify the factors in the different phases of delay. 

Comment 5: 

Literature Review: 

Expand the literature review to include more suboptimal factors in the discussion. 

Avoid singular focus and explore a comprehensive range of factors such as education, financial costs, health insurance, wait times, age, and gender in the context of access to healthcare. 

Ensure clarity and conciseness in presenting the expanded list of factors for a more comprehensive understanding. 

Response 5: 

We understand your concern. We agree that the above-mentioned factors may play a role in access to health care. We decided to mostly focus our discussion on factors identified within the Perinatal Audit registry. Because of the way the registry is constructed and the retrospective nature of the study, we were unable to assess the influence of factors such as education, financial costs, health insurance, wait times, age, and gender on care. Considering your comment, we added women’s educational attainment as a phase 1 and 2 barrier to the discussion. We did not discuss the financial costs of pregnancy care because in the Netherlands these are covered by healthcare insurance (also for refugees and undocumented migrants). Financial costs are therefore considered less of a barrier to accessing care in our country. In addition, there are negligible wait times for accessing antenatal care. As pregnant women are generally around the same age and of the same gender, we did not consider these factors relevant in light of the population we are studying. 

Comment 6: 

Conciseness: 

Make the paper more clear and concise, avoiding unnecessary details. 

List and present factors explicitly to enhance readability. 

By addressing these points, the paper can become more accessible, visually appealing, and comprehensive. 

Response 6: 

We tried to remove unnecessary details by removing table 2, removing unnecessary information from table 3 and by partially rewriting the text. We hope this makes the results more accessible and clearer. 

We kindly thank you again for your input and hope that you consider the revised version more accessible, visually appealing and comprehensive. 

 

Reviewer #2 

Dear reviewer, 

Thank you for your thorough review of our manuscript and your constructive feedback. We used the track changes tool to indicate the edits we made. 

Comment 1: 

The author has written the overall manuscript well. However, the sample size and conclusion are dubious. 

Response 1: 

We understand the reviewer’s concerns considering the sample size. However, since we used the national registry of the perinatal audit and included all cases of refugees, this is the maximum sample size we had access to. We are confident our sample holds sufficient data to provide important qualitative insights regarding care in this specific population. We adjusted the conclusion to better fit the results and the design of the study.

Comment 2: 

Line 209. 

Pakistan and Afghanistan are not Middle Eastern countries. I suggest the authors revise the tables and findings. 

Response 2: 

Thank you for this important comment. We changed the name of the country group from 'Middle East’ to ‘Asia’. 

Comment 3: 

Line 260. 

Table 3. 

Postnatal maternity care should incorporate dietary considerations, including religious restrictions, for refugees. It is pertinent to evaluate why dietary factors are not addressed in table 3 during institutional or non-institutional-based care, particularly given the focus of the study on refugees. The lack of attention to this aspect may lead to suboptimal care and outcomes for refugee women, who may have unique dietary needs due to religious restrictions or other factors. 

Response 3: 

We agree that diet may affect maternal and perinatal outcomes. We were only able to focus on factors identified within the Perinatal Audit registry in this study. Because of the way the registry is constructed and the retrospective nature of the study, we were unable to assess the influence of dietary restrictions. This would make for interesting further research. In general, dietary options in the Netherlands will fulfill people’s needs. Fasting due to Ramadan is usually partially done during pregnancy. Possible effects on fetal growth are indicated by the known birth weights.

Comment 4: 

Line 408-409 

The overall conclusion appears to be too general in nature. Specifically, the authors should direct their attention towards addressing the unique needs of refugee women in the Netherlands. A more targeted approach would be more effective in delivering the intended message. 

Response 4: 

We adjusted the conclusion to better fit the results and the design of the study.

 

Reviewer #3 

Dear reviewer, 

Thank you for your thorough review of our manuscript and your constructive feedback. We used the track changes tool to indicate the edits we made. 

Comment 1: 

In the throes of the world's current multiple wars, within and between many countries, the authors' research is a timely and significant contribution towards global best practice for managing the welfare of refugees, in particular, pregnancy and perinatal care. 

Their data analysis appears appropriate for a reviewer with limited statistical expertise that I am. 

Response 1: 

Thank you for your kind words.

Comment 2: 

However, their findings do not support the generalizability reflected in their results and conclusions, and their "Strengths and Limitations" section partly reveals this point. There are methodological limitations including retrospective design, using a national audit whose cases are incomplete, with an unknown denominator, subjective criteria for case inclusion/rejection by the research team, absence of non-refugee population data for background and comparison. 

In my opinion, 'minor revision' is warranted such that the manuscript would meet acceptance for publication. Its conclusions and recommended policy adjustments ought to be limited to the scope of the audit, namely, the findings that need future corroboration to drive policy change. 

Response 2: 

We agree with the reviewer and acknowledge the limitations of this study. However, we think our results provide valuable data to identify important factors to improve maternal and newborn care for refugees which is partially independent of the data’s completeness. We adjusted the conclusion to better fit the results and the design of the study.

 

Response to reviewers’ comments in text:

Abstract:

Reviewer’s comment:

Clarify your study by explicitly mentioning the retrospective audit design and study period, and briefly noting the sample size (53 cases) for transparency. The audit involves a review of past records and performance itself. How important is it to mention the retrospective audit over here? Additionally, succinctly categorize the severity and types of adverse outcomes to offer readers a clearer understanding of your research context and findings.

Author's response:

We clarified the abstract according to the reviewer’s comment.

Keywords:

Reviewer’s comment:

Choose specific and broad keywords, with synonyms, for clarity and broaden the audience. Limit the list to 5-6 words. Consult the latest PLOS ONE guidelines for keyword count and word limits for the single keyword and consider adding "maternal and newborn health.

Author's response:

We’ve adjusted keywords according to the reviewer’s comment and hope this will broaden the paper’s audience.

Introduction:

Reviewer’s comment :

Improve the flow with transitions, state more about how your study address the challenges and its contribution to resolving the challenges, and incorporate a clear research question to offer the readers clear directions

Author's response:

We strived to improve the flow, clarified the research question, and stated additional information about the goal of the study and how it contributes to improving birth care for asylum seekers and refugees.

Methods:

Reviewer’s comment:

Please rewrite it as the reader could not get sense from it.

Author's response:

We revised both the aforementioned sentence and its precursor.

Reviewer’s comment:

Better to present this model in the framework in the diagrammatic form . Framework is usually looks good in frame.

Author's response:

We added the model in diagrammatic form.

Reviewer’s comment:

How do you determine the factor’s likelihood of contributing to the problem is major or minor?

Author's response:

We clarified that this was based on expert opinion

Reviewer’s comment:

I think its crucial to provide the detailed information regarding the research documentations to ensure the transparency and compliance with the ethical standards within the research.

Author's response:

We added additional information on this topic.

Results:

Reviewer’s comment:

Rewrite it : please never starts with number in the beginning of the paragraph.

Author's response:

We rewrote the paragraph

Reviewer’s comment:

Please make it concise and clear.

Author's response:

We rewrote the paragraph to try to make it more concise and clear.

Reviewer’s comment:

Why the dublicate title in same paper? I suggest you to change the title.

Author's response:

We changed the title.

Reviewer’s comment:

Would you please clear about the what is 12/1 or 13/2. I didn’t get the idea. Rearrange the table in standard format with clear view.

Author's response:

After discussion with all authors, we decided to remove the table entirely as we agree with the reviewer that it was difficult to interpret and added little to no extra information to the article. We adjusted the text accordingly.

Discussion:

Reviewer’s comment:

List more literature review to make your discussion more strong and effective.

Author's response:

We added additional references (which are not marked with track changes as they were added through Mendeley Reference Manager).

Reviewer’s comment:

You have talk about the language barrier and the policy of migrant as a factor. But is it the only factors that is associated for access of services? What about education level, age and gender, health literacy, financial barriers,?? Why you are just focusing these two factors only among a lot??

Author's response:

We understand your concern. We agree that the above-mentioned factors may play a role in access to health care. We decided to mostly focus our discussion on factors identified within the Perinatal Audit registry. Because of the way the registry is constructed and the retrospective nature of the study, we were unable to assess the influence of factors such as education, financial costs, health insurance, wait times, age, and gender on care. Considering your comment, we added women’s educational attainment as a phase 1 and 2 barrier to the discussion. We did not discuss the financial costs of pregnancy care because in the Netherlands these are covered by healthcare insurance (also for refugees and undocumented migrants). Financial costs are therefore considered less of a barrier to accessing care in our country. In addition, there are negligible wait times for accessing antenatal care. As pregnant women are generally around the same age and of the same gender, we did not consider these factors relevant in light of the population we are studying. 

Reviewer’s comment:

You yourself are saying its underrepresentation of the study population so how it gonna represent the other refugees?

Author's response:

What we mean here is that there are probably more factors that play a role in prenatal care for refugees than we were able to identify in this study (due to the design of the National Preinatal Audit and retrospective nature of the study). This makes the high number of suboptimal factors identified even more alarming.

Conclusion:

Reviewer’s comment:

I did not find any new and innovative things over here. These are already published similar result in other publication.

Author's response:

We adjusted the conclusion to better fit the results and the design of the study.

Reviewer’s comment:

I think it is better not to keep in the conclusion part. You can

Author's response:

We moved the further research recommendations to the last paragraph of the discussion. As we considered it important to maintain these recommendations.

---

## [Decision Letter · Decision Letter 1]

5 Jun 2024

Suboptimal factors in maternal and newborn care for refugees: lessons learned from perinatal audits in the Netherlands.

PONE-D-23-31690R1

Dear Dr. Anouk Verschuuren,

We’re pleased to inform you that your manuscript has been judged scientifically suitable for publication and will be formally accepted for publication once it meets all outstanding technical requirements.

Kind regards,

Omid Dadras, MD, PhD

Academic Editor

PLOS ONE

---

## [Editor Report · Acceptance letter]

18 Jun 2024

PONE-D-23-31690R1 

PLOS ONE

Dear Dr. Verschuuren, 

I'm pleased to inform you that your manuscript has been deemed suitable for publication in PLOS ONE. Congratulations! Your manuscript is now being handed over to our production team.

Kind regards, 

on behalf of

Dr Omid Dadras 

Academic Editor

PLOS ONE